# Identification of Key Design Phase-Related Risks in DBB Projects in the UAE—Towards Developing a BIM Solution

Deena Badran [1], Radhi AlZubaidi [1,*] and Senthilkumar Venkatachalam [2]

1   College of Engineering, University of Sharjah, Sharjah P.O. Box 27272, United Arab Emirates; u15105722@sharjah.ac.ae
2   Indian Institute of Technology Palakkad, Palakkad 678 623, India; senthil@iitpkd.ac.in
*   Correspondence: ralzubaidi@sharjah.ac.ae

**Abstract:** Building Information Modelling (BIM) has become a widely used concept in high-rise buildings in the United Arab Emirates (UAE). However, the risks associated with the design phase of multi-story high-rise building projects have, to date, not been addressed by the research studies conducted in the UAE. The results show that "Time Spent in Approval Process" is the main cause of delay, "Complying with the New Regulations and Rules" is the main cause of cost overrun and "Poor Coordination between the Design Disciplines" is the main cause of the quality deficiency. The results also indicate that "Poor Coordination between the Design Disciplines" is the only risk that has a high-risk index related to time, cost, and quality. "Change Initiated by the Client" and "Interface by the Client during the Design Process" are ranked among the top five risks which cause delay and cost overrun in UAE high-rise buildings. The paper mapped the root causes of the identified most significant risks against the possible BIM-based solutions. The results show that BIM can effectively mitigate 75% of the root causes of these risks and, further, BIM is also effective in managing the consequences of the root causes for the remaining the remaining 25%.

**Keywords:** risk management; building information modelling; design–bid–build; design phase; high-rise buildings





## 1. Introduction

In recent years, the construction sector in the United Arab Emirates (UAE) has experienced consistent growth in the construction sector, which is evidenced in its contribution to the country's Gross Domestic Product (GDP). One of these challenges is managing project risks. The design stage was found to have a significant effect on managing the risks during the project's lifecycle [1–5]. However, the implementation of risk management during the design stage has faced difficulties arising, due to the fact that the modern design phase has numerous interdependent, knowledge-intensive multidisciplinary tasks, and the overall process is inherently iterative [6,7]. Currently, BIM has been adopted to provide a platform for coordinating between multidisciplinary teams [8,9]. The construction industry has benefited in many areas through BIM. As a result, it would be natural to investigate its capabilities in managing factors, such as time, cost, and quality-related risks, during the design phase, as stated by [1]. Recent research recommends the inclusion of all possible risks faced by the projects to achieve a comprehensive BIM-based risk management solution as the risks vary depending on the project's type and level of complexity, and the BIM levels required to manage these risks vary accordingly [10–13]. However, few studies have been conducted in the UAE to determine and assess the risks associated with the design phase of the construction projects delivered under design–bid–build (DBB) contracts, especially for high-rise buildings [14–16]. The aim of this research is to develop a BIM-based risk management solution in the design stage for design–bid–build (DBB) high-rise building projects in UAE by (1) identifying the key design phase to related risks

affecting the construction objectives of DBB projects in the UAE, (2) identifying the root cause of the most significant risks, and (3) mapping the root causes of the most significant risks against possible BIM-based solutions.

The research paper contributes to the literature by identifying the key design phase-related to risks that have an impact on the time, cost and quality aspects of high-rise building projects delivered under the design–bid–build (DBB) method. This will assist in developing a BIM-based risk management framework applicable for managing the design phase-related risks that have an impact on time, cost, and quality aspects of high-rise building projects. Furthermore, the research will contribute to the industry by assisting in developing a framework based on the real solution risks and determining causes identified from high-rise building projects in the UAE construction market. The research was conducted within the following scope (as illustrated in Figure 1 with shaded annotations): (1) design phase of projects; (2) projects delivered under the DBB delivery method; (3) risk impacts related to time, cost, and quality; (4) private projects; and (5) high-rise building projects. The research was conducted in four emirates within the United Arab Emirates.

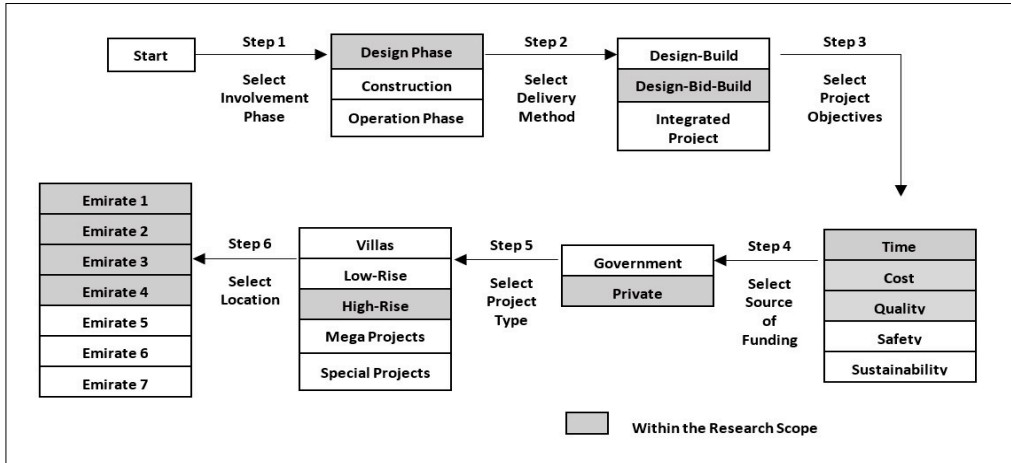

**Figure 1.** Research scope.

As a result of its unique features, any construction project may be subject to more risks than many other industries [17–19]. These risks normally affect one or more of the projects aims, including time, cost, and quality.

Previous studies have shown that the expected risks should be managed at an early stage in the project life cycle [1–5]. It is evident that ignoring risks during the design phase negatively affects progress in the subsequent phases. Furthermore, the success of project completion will be affected adversely as risks amplify over time [20]. The UK's Construction Design and Management Regulation in 2007 emphasized the role of the design phase in risk management and showed that risk management should be applied to the lifecycle of the project, starting from the design stage phase, and ending with demolition time.

## 2. Literature Review

A construction project is a one-off endeavor. It has many unique characteristics, such as a long period, complex processes, abominable environment features, financial intensity effects and dynamic organization structures [21].

The literature review conducted by [22] revealed that design phase-related risks have a distinctive significant impact on construction projects. Their critical review showed that any change in the design by the owner occurring during the construction phase was most likely to cause a delay in the time of construction projects in Saudi Arabia [23]. Design-related risk, including incomplete drawings and changes in design, was one of the most critical risks in the construction sectors in Sri Lanka [24], and one of the five most factors that showed significant risks in the construction sector in China [25]. Poor design ability and

mostly design changes were the third most critical factor causing failure in Vietnamese construction projects [26], number four was design inefficiencies [27]. Mistakes or omissions in consultant documents was the greatest risk factors that affected the budget and quality of construction projects in Denmark [28].

The implementation of risk management during the design stage is faced with challenges created by the modern design process, which has numerous interdependent, knowledge-intensive, multidisciplinary tasks that are iterative in nature occurrence [6,7]. Thus, information exchange management forms the basis for managing risks. There are many frameworks, tools and methods have been developed and utilized around the world, such as Design Structure Matrix (DSM)-based methods [6,7,29], and Multi Domain Mapping Method (MDM) [8,9,30]. However, only Building Information Modelling is widely used by [31]. The benefits gained from using Building Information Modelling in communication and collaboration in the design phase increased the expectation that it can play a distinctive role in managing project risks [1,32].

BIM is defined in different ways. All the definitions, however, concentrate on storing and managing data produced throughout the project lifecycle, integrally, to produce the 3D model, 4D time simulation, 5D cost budget, 6D facility management plan, 7D sustainable design, and 8D safety management plan. Building Information Modelling is not a tool, it represents a data management platform which requires solid integration of technologies, people, and processes; [33] defined Building Information Modelling as a set of interacting policies, processes and technologies used in creating a "methodology to control the most important building design and project data in digital format throughout the building's life cycle [31]. Building Information Modelling can add an infinite number of dimensions to the Building Model [34].

Recent research trends of BIM show the need for the identification of possible risks faced by the projects to modify a well-comprehensive BIM-based risk management solution. In their study [10] concluded the inclusion of a more comprehensive risks list for further development of their BIM-based risk management methodology. Furthermore, the researchers recommended validating their methodology on construction projects with different types, levels of complexity, and demands. Ref. [13] developed a BIM-based risk management framework that can be applied only to internal risks which can be responded to by BIM. The risks used to develop the framework were identified from the literature and verified by BIM experts. Ref. [12] emphasized the need to establish a risk database to link it with different levels of BIM to support a BIM-based risk management approach. Ref. [35] depended on the literature review to identify and evaluate the risks used to propose a methodology for BIM-based risk management. Ref. [11] concluded through two case studies, Central Information Resource Centre and Multistorey Commercial Building, that most of the identified high-ranked risks were either omitted or addressed, reducing their impacts considerably. However, this conclusion requires further investigation as the risks vary depending on the project's type and level of complexity. The same was highlighted by the researchers who found that the rank of risks and their impact on project objectives vary between the two case studies. Furthermore, the systematic risk management methodology proposed by [11] showed that the knowledge of the risks faced by the projects is the base for any successful BIM-based solution. They studied the new risks that emerged from the implementation of BIM and found that a 'lack of BIM knowledge' significantly impacted the project objectives. This new consideration of implementing BIM in risk management has emphasized the importance of identifying the real risks impacting the projects, to ensure that the level of BIM required to manage these risks is achievable.

The literature review revealed that no study has been conducted to date to determine the key design phase-related risks that have a distinctive effect on the time, cost, and quality aspects of the construction sectors in the UAE. Ref. [15] surveyed to determine and assess the most frequent causes of claims in road construction projects sectors in the UAE in addition to the root causes that assist in the occurrence of claims. Ref. [16] adopted an in-depth interview study involving clients, contractors, and consultants to

study different risks and risk factors related to the success of UAE construction sector projects. Ref. [36] conducted a concurrent mixed-methods approach, using a questionnaire and an interview with UAE construction experts, to identify the major causes of poor time and cost performance. Ref. [37] conducted a questionnaire survey study among construction experts in the UAE to analyze the types and causes of construction claims and their frequency and severity. Ref. [38] conducted a face-to-face questionnaire survey among construction professionals in Abu Dhabi to determine the key risks causing delays in construction projects in Abu Dhabi city in the UAE. Ref. [39] investigated the delay in construction project sectors in the UAE market.

This lack of information on the design phase-related to risks requires an effort to fill the present gaps in the literature to enable the process of developing a BIM-based risk management solution applicable for the design phase stage of the construction projects in the UAE, specifically for the high-rise building projects, to control the key risks that affecting the time, the cost, and the quality aspects of these projects [40].

### 3. Research Methodology

The research methodology contained three steps as shown in Figure 1. The first was to determine the key design phase that related to risks affecting the time, cost and quality of projects delivered under a design–bid–build contract. A total of 37 design-related risks, shown in Table 1, were identified through the literature review [12,21,22,27,41–49] and interviews with experts in the construction sectors in the UAE. A set of questionnaires were developed to evaluate perceptions of the likely occurrence of these risks and their effects on time, cost, and quality. The second step was to investigate the root causes of the identified key design phase that related to risks. The third was to map the root causes of the most distinctive high-ranked risks against a possible BIM-based solution.

**Table 1.** List of design phases that related to risks.

| No. | Category | ID | Design-Phase Related Risks |
|:---:|:---:|:---:|:---:|
| | | | **Risks from Literature Review** |
| 1 | Client | ICD | Interference by the client in the design process |
| 2 | Client | CIV | Client-initiated modifications/client requests changes in the design |
| 3 | Client | DRD | Client delays in reviewing and approving design |
| 4 | Client | UCR | Clients' unrealistic initial requirements and unreasonably high expectations |
| 5 | Client | UCD | Unrealistic contract duration imposed by client/pressure to deliver design in an accelerated schedule |
| 6 | Client | DPP | Delay in progress payments by the client |
| 7 | Client | FCC | Financial constraints faced by the client |
| 8 | Client | DPS | Design process suspended by the client |
| 9 | Consultant | DPD | Delay in preparation of drawings |
| 10 | Consultant | AMD | Ambiguities/imperfections/mistakes in drawings and specifications |
| 11 | Consultant | PIQ | Poor implementation of quality control/assurance (QC/QA) |
| 12 | Consultant | PCC | Poor communication between consultant and other project parties |
| 13 | Consultant | IDC | Insufficient data collection and survey before the design |
| 14 | Consultant | MID | Mistakes in the design |

**Table 1.** *Cont.*

| No. | Category | ID | Design-Phase Related Risks |
|---|---|---|---|
| 15 | Consultant | SMS | Shortages in the materials specified by the consultant/materials required approval from the concerned authorities |
| 16 | Approval Authorities | TSA | Time spent in the approval process |
| 17 | Approval Authorities | CRL | Changes in government regulations and laws |
| 18 | Other | UGC | Unforeseen ground conditions (such as unexpected geotechnical or groundwater issues, underground utility lines) |
| 19 | Other | DCE | Deficiencies or inaccuracies in cost estimation |
| 20 | Other | DPS | Deficiencies in planning and scheduling the project |
| 21 | Other | IOS | Inappropriate overall organizational structure of companies linked to the project |
| 22 | Other | MCD | Mistakes and discrepancies in the contract documents |
| 23 | Other | LDP | Legal disputes between various parties in the project |
| **Risks from Pilot Study** | | | |
| 24 | Client | JOP | Joint ownership of project |
| 25 | Client | NSE | New stakeholder emerges and requests changes |
| 26 | Client | CEU | Change in the end users |
| 27 | Consultant | MCR | Misunderstanding of client requirements |
| 28 | Consultant | PCD | Poor coordination between design disciplines |
| 29 | Consultant | IDC | Impractical design/constructability issues not studied during the design phase |
| 30 | Consultant | DMP | Difficulty in measuring progress during design development |
| 31 | Consultant | ITR | Insufficient time to review tender documents submitted by contractor |
| 32 | Consultant | MIB | Mistakes and discrepancies in the itemized bill of quantities (BOQ) prepared by the cost consultant |
| 33 | Consultant | RCF | Reduction in consultant fees |
| 34 | Consultant | CSC | Client changes the consultant |
| 35 | Other | ADU | As-built drawings are not available for the existing structures |
| 36 | Other | SIF | Slow information flow between project team |
| 37 | Other | APM | Absence of professional project management |

*3.1. Research Step 1*

A study conducted face-to-face interviews with clients, senior engineering consultants, and project management consultancies, and the frequency of meetings. They were asked to rank the risks collected from the previous studies and pilot studies (Figure 2) and identify the top 10 risks that affect the time, cost, and quality of the high-rise DBB building project sectors in the UAE.

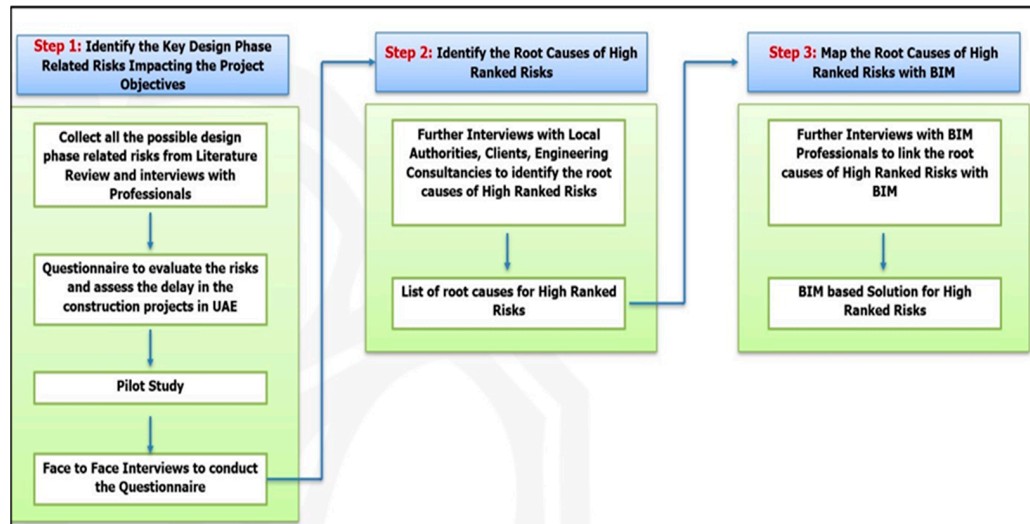

**Figure 2.** List of design phases related to risk impact, time, cost, and quality.

The interview was structured into three parts. The first section sought general information about the respondent's background to ensure that the response fulfils the criteria of the scope of the research shown in Figure 3. The second part included respondents asking about delay causes and cost overruns in high-rise building project sectors in the UAE, and the involvement of each project's stakeholders in such delays and cost overruns. In addition, participants provided information about the role that the design phase plays in the risk management phase process. In the third part of the interview, respondents have been asked to rank the likelihood and impact of each risk. A five-point scale, proposed by [49], was used for ranking. The assessment of impact was based on a scale of 1 = very low (0–10%), 2 = low (10–30%), 3 = moderate (30–50%), 4 = high (50–70%), 5 = very high (70–100%). The assessment of the likelihood of occurrence was based on a scale of 1 = very unlikely (0–10%), 2 = unlikely (10–30%), 3 = moderate (30–50%), 4 = likely (50–70%), 5 = very likely (70–100%). The scale used in assessing the likelihood of occurrence and impact of risk was explained to the respondent before the start of the interview.

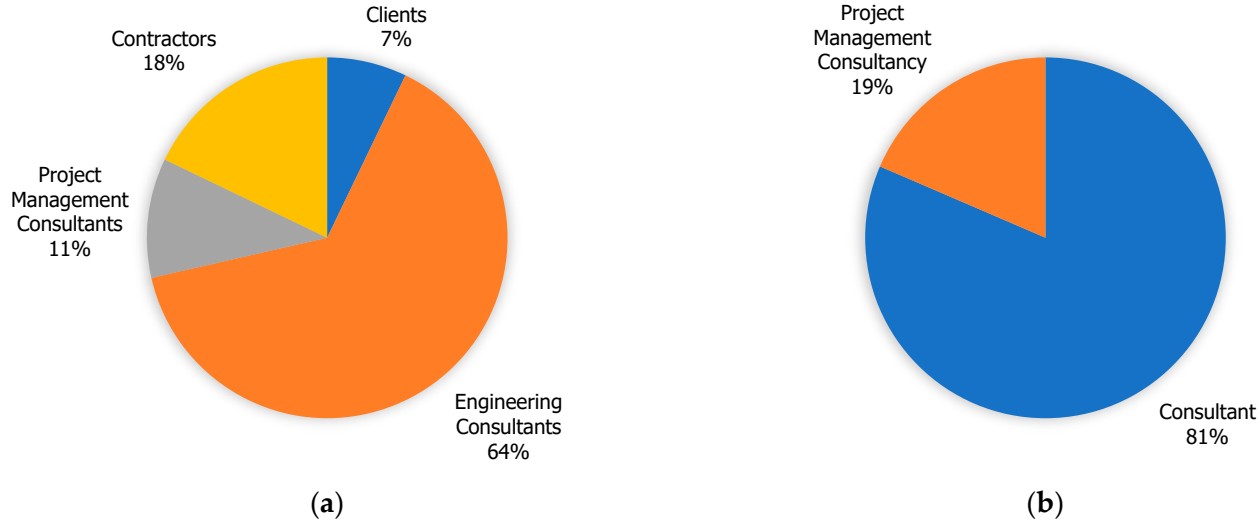

**(a)**  **(b)**

**Figure 3.** *Cont.*

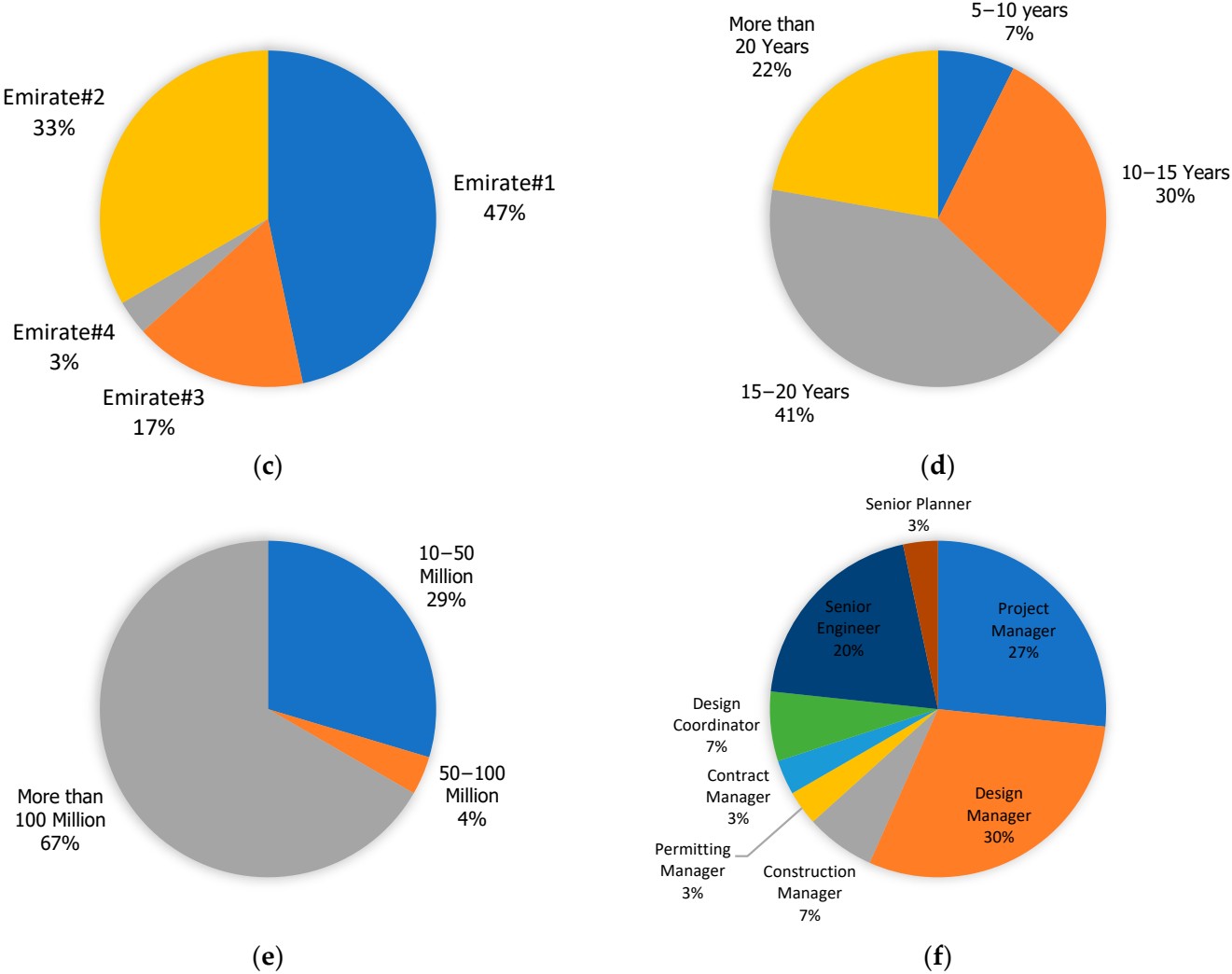

**Figure 3.** Distribution of respondents (interviews). (**a**) Distribution of Respondents by Profession (Pilot Study); (**b**) Distribution of Respondents by Profession (Interviews); (**c**) Distribution of Respondents by Location (Interviews); (**d**) Distribution of Respondents by Experience (Interviews); (**e**) Distribution of Respondents by Maximum Project Value Involved (Interviews); (**f**) Distribution of Respondents by Job Position (Interviews).

Stratified random sampling was adopted to ensure that the group was representative of the construction business in the UAE. The number of respondents from each emirate was calculated based on the number of companies eligible to work in high-rise buildings in that part of the country. The data were collected from the economic department and municipality in each emirate, as shown in Table 2. There were 30 respondents. The stratified sample size was calculated, as shown in Table 3.

**Table 2.** Population (Number of Companies Eligible to Work in High-Rise Building Projects).

| Emirate | Engineering Consultancies | Project Management Consultancies | Total Number of Companies |
|---|---|---|---|
| Emirate 1 | 273 | 17 | 290 |
| Emirate 2 | 193 | 27 | 220 |
| Emirate 3 | 96 | 8 | 104 |
| Emirate 4 | 5 | 0 | 5 |
| Total | | | 619 |

**Table 3.** Adopted Stratified Sampling in the Present Research.

| Emirate (Strata) | Strata Sample Size | Division of Strata Sample | | |
| --- | --- | --- | --- | --- |
| | | Engineering Consultancies | Project Management Consultancies | Total Number of Companies |
| Emirate 1 | 14.1 | 13.2 | 0.8 | 14.0 |
| Emirate 2 | 10.7 | 9.4 | 1.1 | 10.5 |
| Emirate 3 | 5.0 | 4.7 | 0.4 | 5.0 |
| Emirate 4 | 0.2 | 0.2 | 0.0 | 0.2 |
| | Total | | | 29.7 |

The face-to-face interview methodology has been selected for the following reasons: (1) to increase the level of confidence in collected data by ensuring that all respondents have received the same level of understanding, (2) to investigate the root causes of the highest-ranked risks, and (3) investigate the possibility to have more risks other than the collection from the literature review.

A pilot study was conducted with 20 respondents, including engineering consultants, project management consultants, and owners. All the comments provided by the respondents in the interviews were analyzed. All the variables proposed by the respondents were studied and implemented as required. Some of the new risks suggested by the respondents were considered in the study. The risks index of identified design phase-related risks is shown in Table 4.

**Table 4.** Risks index of identified design phase-related risks.

| Risk ID | Time Delay | | Cost Overrun | | Quality | |
| --- | --- | --- | --- | --- | --- | --- |
| | Risk Index | Rank | Risk Index | Rank | Risk Index | Rank |
| ICD | 0.457 | 27 | 0.337 | 28 | 0.162 | 17 |
| CIV | 0.447 | 1 | 0.393 | 2 | 0.191 | 15 |
| DRD | 0.305 | 2 | 0.169 | 17 | 0.059 | 35 |
| UCR | 0.190 | 28 | 0.166 | 1 | 0.129 | 8 |
| JOP | 0.295 | 17 | 0.229 | 32 | 0.179 | 34 |
| UCD | 0.165 | 9 | 0.203 | 27 | 0.148 | 14 |
| DPP | 0.247 | 3 | 0.155 | 14 | 0.070 | 2 |
| FCC | 0.224 | 5 | 0.134 | 15 | 0.217 | 32 |
| DPS | 0.311 | 15 | 0.230 | 9 | 0.091 | 5 |
| NSE | 0.044 | 14 | 0.044 | 5 | 0.023 | 1 |
| CEU | 0.124 | 35 | 0.121 | 29 | 0.056 | 6 |
| MCR | 0.193 | 33 | 0.176 | 33 | 0.138 | 28 |
| DPD | 0.242 | 7 | 0.123 | 35 | 0.047 | 19 |
| AMD | 0.272 | 13 | 0.260 | 6 | 0.196 | 12 |
| PIQ | 0.273 | 32 | 0.250 | 18 | 0.299 | 16 |
| PCC | 0.184 | 29 | 0.108 | 12 | 0.132 | 24 |
| PCD | 0.390 | 8 | 0.354 | 19 | 0.329 | 4 |
| IDC | 0.214 | 31 | 0.197 | 3 | 0.120 | 18 |
| IDC | 0.173 | 18 | 0.175 | 4 | 0.139 | 20 |
| MID | 0.152 | 34 | 0.145 | 34 | 0.119 | 22 |
| DMP | 0.043 | 12 | 0.035 | 7 | 0.038 | 25 |
| ITR | 0.109 | 4 | 0.118 | 20 | 0.118 | 33 |
| MIB | 0.041 | 16 | 0.037 | 24 | 0.027 | 31 |
| SMS | 0.149 | 19 | 0.140 | 8 | 0.130 | 27 |
| RCF | 0.124 | 6 | 0.111 | 13 | 0.116 | 9 |
| CSC | 0.097 | 20 | 0.091 | 11 | 0.045 | 7 |
| EAP | 0.465 | 24 | 0.265 | 22 | 0.110 | 29 |

**Table 4.** *Cont.*

| Risk ID | Time Delay | | Cost Overrun | | Quality | |
|---|---|---|---|---|---|---|
| | Risk Index | Rank | Risk Index | Rank | Risk Index | Rank |
| CGR | 0.424 | 37 | 0.417 | 37 | 0.148 | 3 |
| UGC | 0.228 | 11 | 0.220 | 25 | 0.060 | 11 |
| ADU | 0.045 | 25 | 0.041 | 16 | 0.045 | 36 |
| SIF | 0.217 | 22 | 0.097 | 31 | 0.113 | 13 |
| DCE | 0.237 | 26 | 0.296 | 26 | 0.181 | 30 |
| DPS | 0.254 | 36 | 0.204 | 36 | 0.115 | 26 |
| IOS | 0.214 | 30 | 0.159 | 10 | 0.215 | 21 |
| APM | 0.270 | 10 | 0.203 | 30 | 0.244 | 37 |
| MCD | 0.072 | 21 | 0.075 | 23 | 0.054 | 23 |
| LDP | 0.125 | 23 | 0.114 | 21 | 0.035 | 10 |

*3.2. Research Step 2 and 3*

After the identification of the key design phase-related risks was complete, an initial exercise was carried out with the BIM experts to map the high-ranked identified risks with the BIM solution and investigate the capability of BIM to manage these risks at an early stage of the project's lifecycle. However, the initial exercise revealed that the assessment of BIM capability to determine risks requires further investigation to identify the underlying causes of the most distinctive risks, which will enable the link between the risks and the BIM uses and contribute to building a BIM-based solution applicable for managing the key risks. This approach was found to be effective. Therefore, more interviews were conducted with senior engineering consultants, clients, local authorities, project management consultants, and BIM professionals to determine the distinctive causes of risks and integrate them with the Building Information Modelling design. Much time and effort were spent setting meetings with key personnel in local authorities. However, their input in identifying and analyzing the root causes of some risks was considered essential for better understanding and linking the BIM uses. Furthermore, the BIM professionals selected for the interviews have significant experience in high-rise building projects and working in management positions, so they are familiar with BIM applications in all design disciplines.

**4. Questionnaire Analysis Approach**

There were two categories of measures. The first was the likelihood of occurrence of each risk, and the second was its level of impact on time, cost, and quality. Based on these measures, the significance index for each risk, as assessed by a respondent, was calculated through Equation (1), as adopted by [49]:

$$r_{ij}^{k} = \alpha_{ij}\beta_{ij} \tag{1}$$

where $r_{ij}^{k}$ = significance index assessed by respondent *j* for the impact of risk *i* on project objective *k*.

*i* = ordinal number of risks.
$i \in (1, m)$; m = total number of risks.
*k* = ordinal number of the project objective.
$k \in (1, 5)$.
*j* = ordinal number of valid feedbacks to risk *i*.
$j \in (1, n)$; = total number of valid feedbacks to risk *i*.
$\alpha_{ij}$ = likelihood occurrence of risk *i*, assessed by respondent *j*.
$\beta_{ij}$ = level of the consequence of risk *i* on project objective *k*, assessed by respondent *j*.

The average score for each risk, considering its significance on a project objective, was calculated through Equation (2) [49].

This average score is called the risk significance index score and was used to rank all risks:

$$\text{Risk Significance Index Score} = R_i^k = \frac{1}{n}\sum_{j=1}^{n}\alpha_{ij}\beta_{ij} \tag{2}$$

Cronbach's $\alpha$ coefficients for risk significance index score for the risks impacting time, cost and quality were 0.83, 0.84 and 0.86, respectively. The reliability of the questionnaire was proven because an $\alpha$ value equal to or greater than 0.70 is considered satisfactory [14].

## 5. Results and Discussion

The distribution of the respondents as per profession, location, experience, maximum project value and job position is shown in Figure 3. The results reveal that 59% of the respondents had experienced delays in more than 50% of the projects in which they were involved. The respondents attributed 40% of the causes of delay to the client and 24% to the consultant. Moreover, the results showed that 60% of the respondents had experienced cost overruns in more than 50% of the projects in which they were involved. The respondents attributed 45% of the causes of cost overrun to the client, and 19% to the consultant. The respondents did not find significant quality deficiencies in the projects. Those that were noted were attributed to the consultant. The investigation revealed that most of the risks by the client and consultant are related to the design phase and can be managed during the early stage of the project's lifecycle.

### 5.1. Risk Significance Index Score

The results in Figure 4 show that 'Poor Coordination between Design Disciplines-PCD' is the only risk which has a high-risk index related to time, cost, and quality, and the values obtained are 0.39, 0.35, and 0.33, respectively. The highest ranked risks are presented from Figures 5–7. The results in Figure 5 reveal that the high-ranked risks associated with the design phase and causing a delay in the projects are ordered from highest to lowest as follows: time spent in the approval process, interference by the client in the design stage process, the client initiated modifications/client request for changes in the project design, changes in government regulations and laws, poor coordination between design disciplines, deficiencies in planning and scheduling the project, client delays in reviewing and approving the design, joint ownership of the project, poor in the implementation of quality control/assurance (QC/QA), and ambiguities/imperfections/mistakes in drawings and specifications.

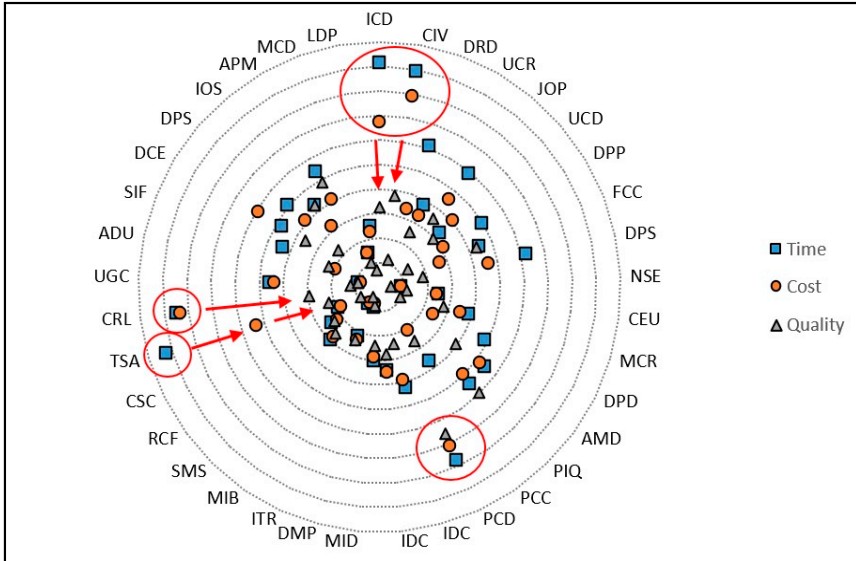

**Figure 4.** Risk significance index score for the design phase-related risks in the DBB process of high-rise building projects in the UAE.

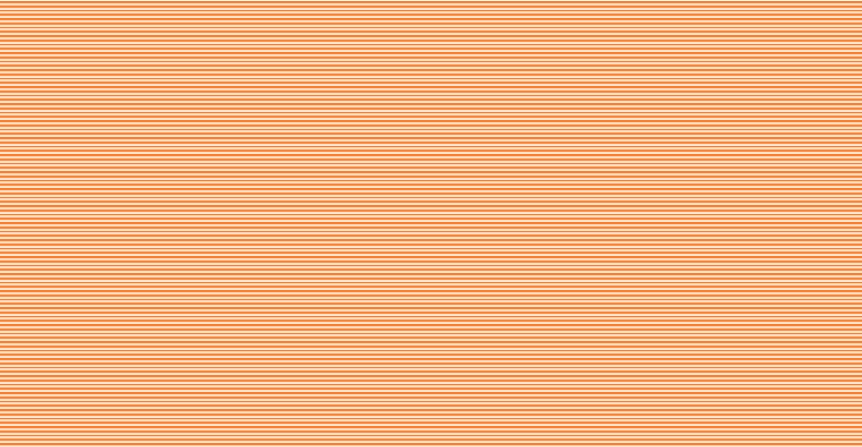

**Figure 5.** Key design phase-related risks in the DBB process affecting the duration of high-rise building projects in the UAE.

The results in Figure 6 reveal that the high-ranked risks associated with the design phase and causing cost overruns in the projects are (ordered from highest to lowest): changes in government regulations and laws, client-initiated modifications/client requests for changes in the design, poor coordination between design disciplines, interference by the client in the design process, deficiencies or inaccuracies in cost estimation, time spent in the approval process, ambiguities/imperfections/mistakes in drawings and specifications, poor implementation of quality control/assurance (QC/QA), deficiencies in planning and scheduling the project, and joint ownership of the project.

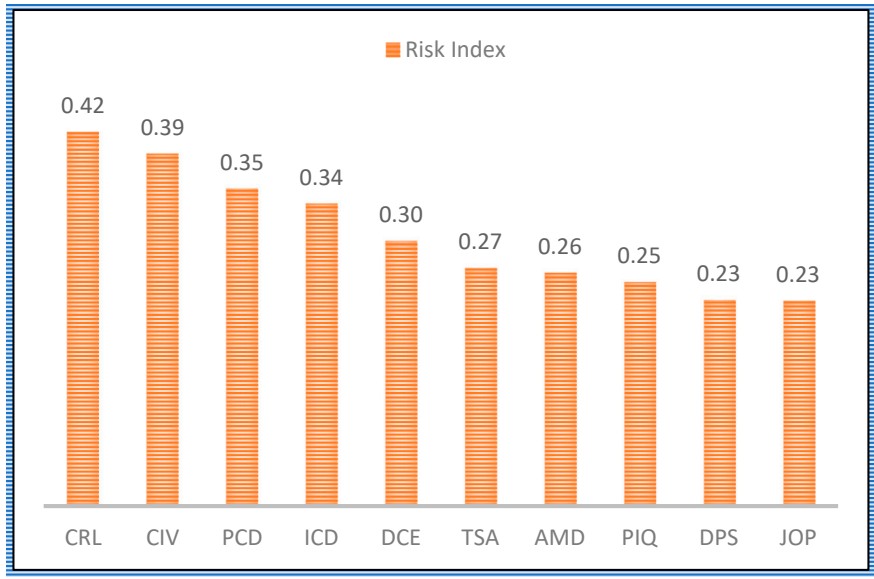

**Figure 6.** Key design phase-related risks in the DBB process affecting the cost of high-rise building projects in the UAE.

The results in Figure 7 reveal that the high-ranked risks associated with the design phase and impacting the quality of the projects are (ordered from highest to lowest); poor coordination between design disciplines, poor implementation of quality control/assurance (QC/QA), absence of professional project management, financial constraints faced by the client, the inappropriate overall organizational structure of companies linked to the project, ambiguities/imperfections/mistakes in drawings and specifications, client-initiated modifications/client request for changes in the design, deficiencies or inaccuracies in cost estimation, joint ownership of the project, and interference by the client in the design process.

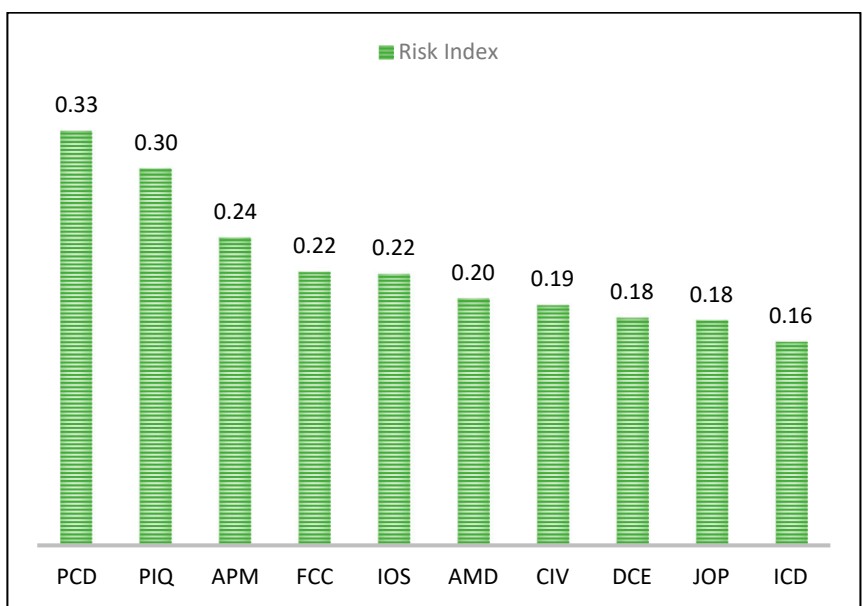

**Figure 7.** Key design phase-related risks in the DBB process affecting the quality of high-rise building projects in the UAE.

The radar chart in Figure 8 shows 'Poor Coordination between Design Discipline.

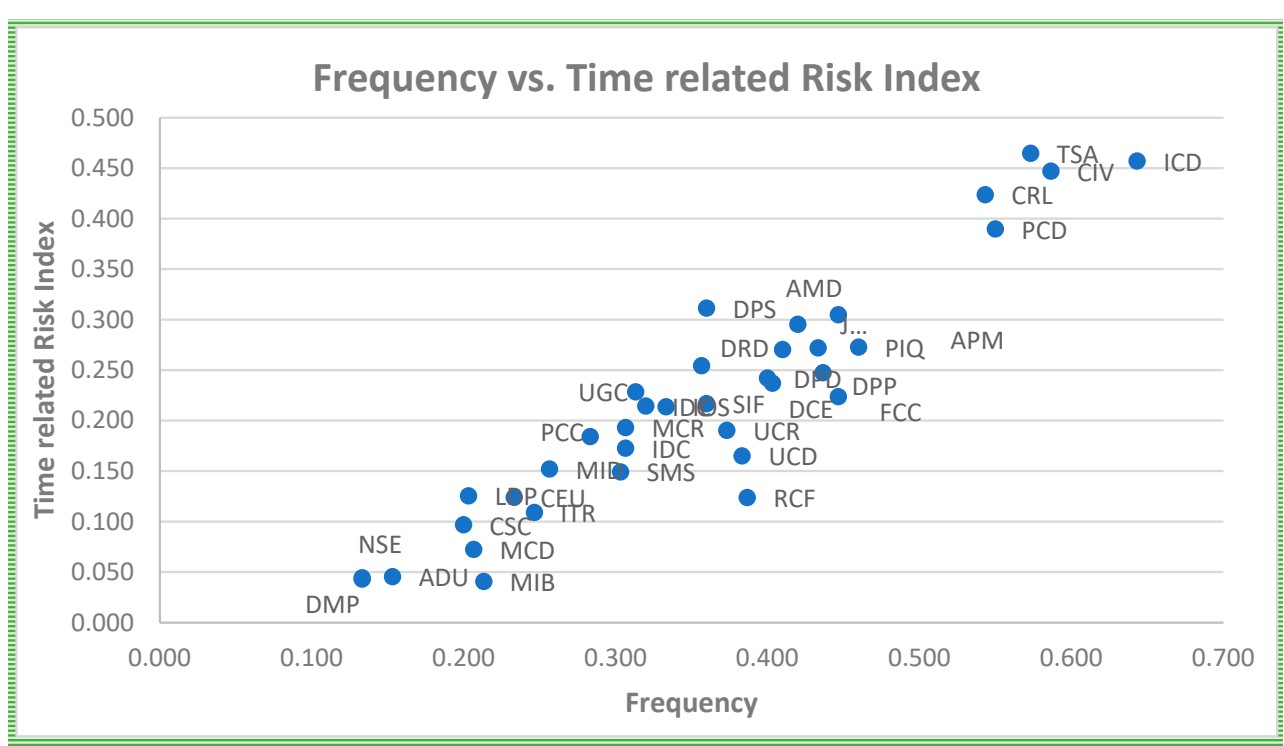

**Figure 8.** Frequency of risk vs. time-related risk index.

Disciplines-PCD is the only risk that has a high-risk index related to time, cost, and quality; the values obtained are 0.39, 0.35, and 0.33, respectively.

The results further show that 'Poor Coordination between Design Disciplines-PCD' has the highest risk index related to quality among all the risks considered in this study. The respondents were asked to evaluate the level of coordination between the design disciplines.

Only 37% of the respondents found the level of coordination to be high (exceeding 70%). The respondents said that miscoordination occurs between all the design disciplines, but the highest risk comes from mechanical, electrical, and plumbing (MEP) with other disciplines. The main causes of this risk were further investigated; the results revealed that these causes were the traditional tools used to coordinate the design disciplines and a change in the design.

The radar chart in Figure 8 shows that 'Change Initiated by the Client-CIV' and 'Interface by the Client during the Design Process-ICD' are ranked among the top five risks causing delay and cost overruns, with the risk index equal to 0.45 and 0.47 for time delay, respectively, and 0.39 and 0.34 for cost overrun, respectively. Most of the interviews conducted with professionals showed that the impact would be amplified when risks occur during the construction phase. However, the results show that the risk index on quality is not high. Further investigation was performed with the project clients to understand the root causes. The investigations showed that the main cause of these risks is that clients are not able to visualize the project or explain their requirements. The clients asserted that their interference and changes are aimed only at improving the project quality.

The radar chart in Figure 8 shows that 'Time spent in the Approval Process-TSA' is the main design phase-related risk that causes delays in DBB high-rise building projects. Further interviews were conducted with the authorities, including municipal, civil defense, and services authorities, to identify the key causes of the delays. The results showed that poor implementation of the QA/QC system by the consultant, discrepancies in drawings, mistakes in design and drawings, and poor implementation of new regulations mainly cause delays in the approval process. The results reveal that this risk does not cause cost overruns in the project and improves the quality of the project. 'Changes in Government Authorities Regulations and Laws' was found to be one of the main causes of delay and cost overruns. However, the results showed that it improves the quality of the project. Further interviews with the approval authorities revealed that when regulations are amended, they affect the projects for a short period until the engineering consultants become familiar with the new regulations.

The relationship presented in Figures 8–11 shows that 'Interface by the Client during the Design Process-ICD' is the risk that has the highest likelihood of occurrence among all the top-ranked time- and cost-related risks. On the other hand, the results show that 'Deficiencies in planning and scheduling of the project-DPS' is the risk that has the highest impact on project duration among all top-ranked time-related risks. The results show that 'Deficiencies or inaccuracy in cost estimation' is the risk that has the highest impact on project cost among all the top-ranked cost-related risks.

The relationship presented in Figures 12 and 13 further show that 'Poor Coordination between Design Disciplines-PCD' is the risk that has the highest likelihood of occurrence and impact on quality among all the top-ranked quality-related risks.

The results show that 59% of the respondents experienced delays in over 50% of the projects in which they were involved. The respondents attributed 40% of the causes of the delays to the client and 24% to the consultant. Moreover, the results showed that 60% of the respondents experienced cost overruns in over 50% of the projects in which they were involved. The respondents attributed 45% of the causes of cost overruns to the client and 19% to the consultant.

Deficiency in quality was found to be minimal in the projects, and the consultant played a significant role in the causes of deficiency. The investigation revealed that most of the causes by the client and consultant were related to the design phase and could be managed during the early stage of the project lifecycle.

The results obtained from the questionnaire were benchmarked with the key risks identified in construction projects in different countries. The findings obtained from the current study were in line with other research findings, though the majority of this research was carried out during the construction stage of the project's lifecycle and on different types of projects. This gives a strong indication that the BIM-based risk management

framework that will be developed based on these results can be implemented in a wide range of projects and other countries.

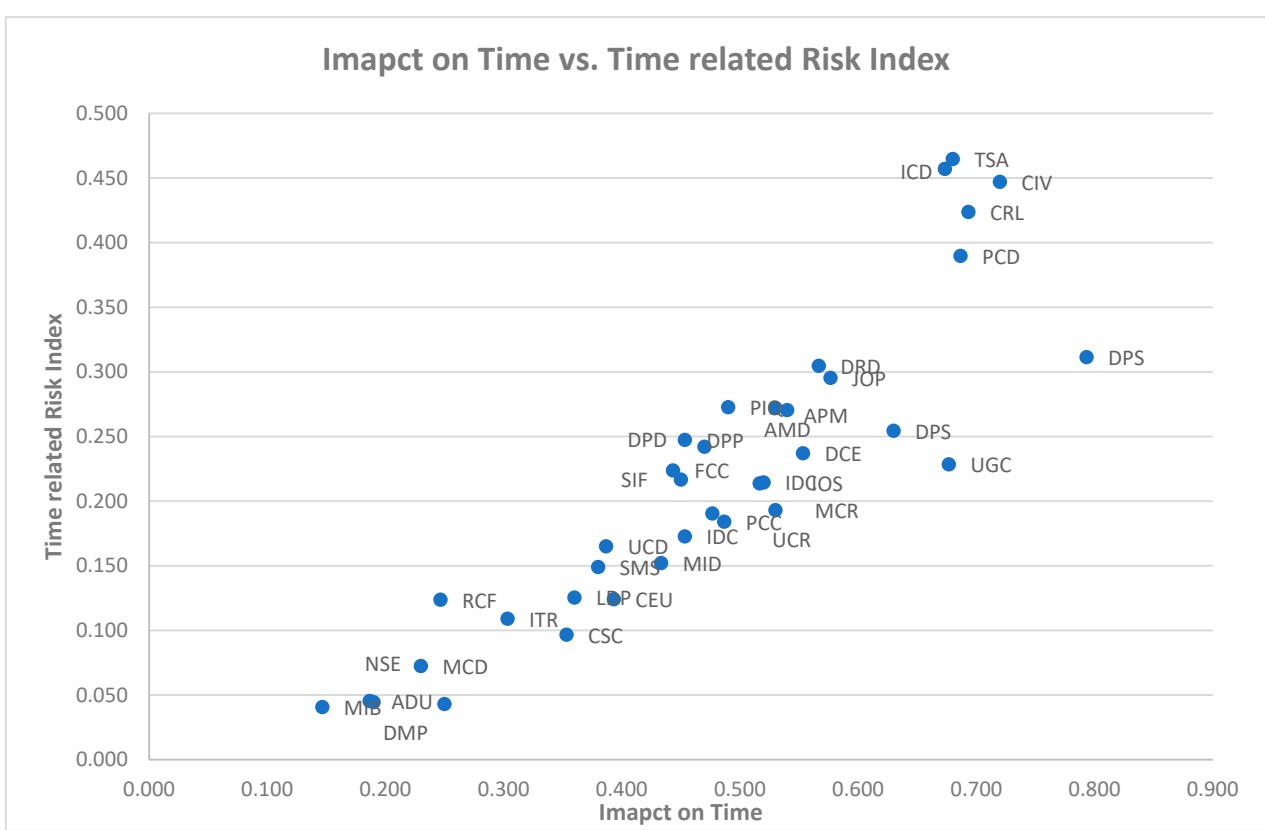

**Figure 9.** Impact of risk on time vs. time-related risk index.

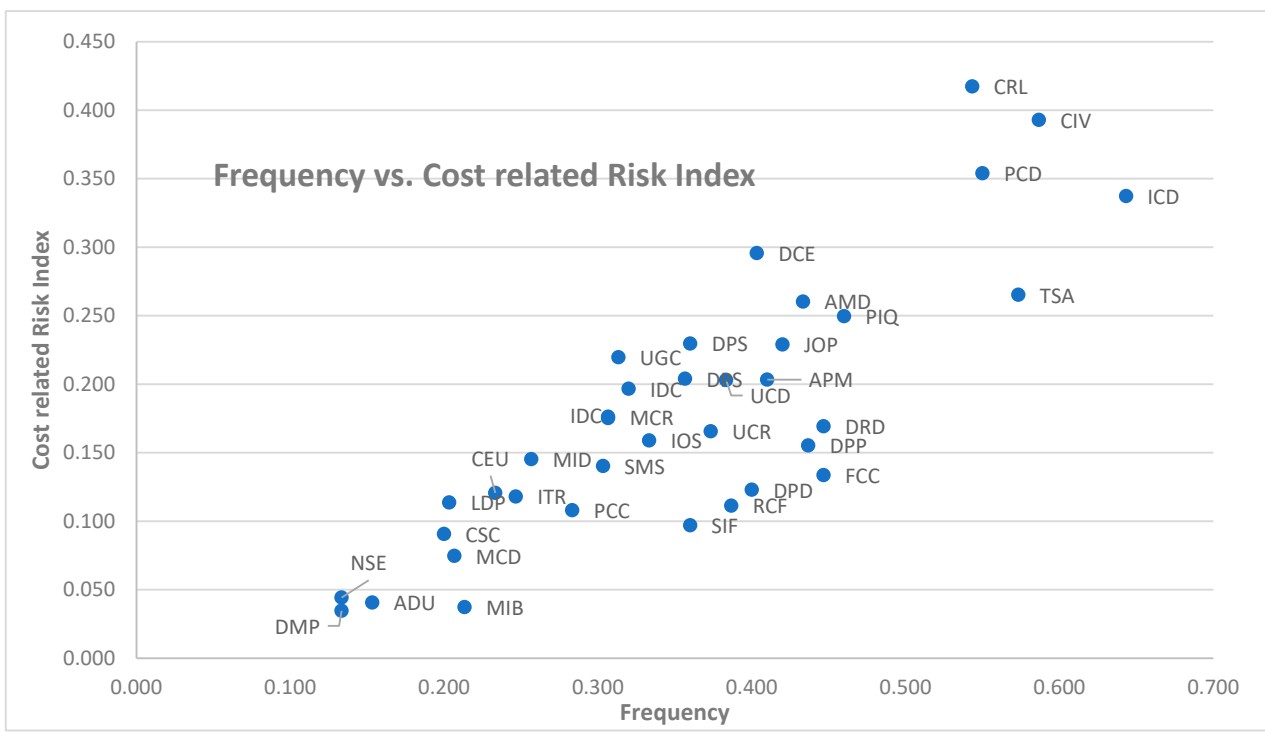

**Figure 10.** Frequency of risk vs. cost-related risk index.

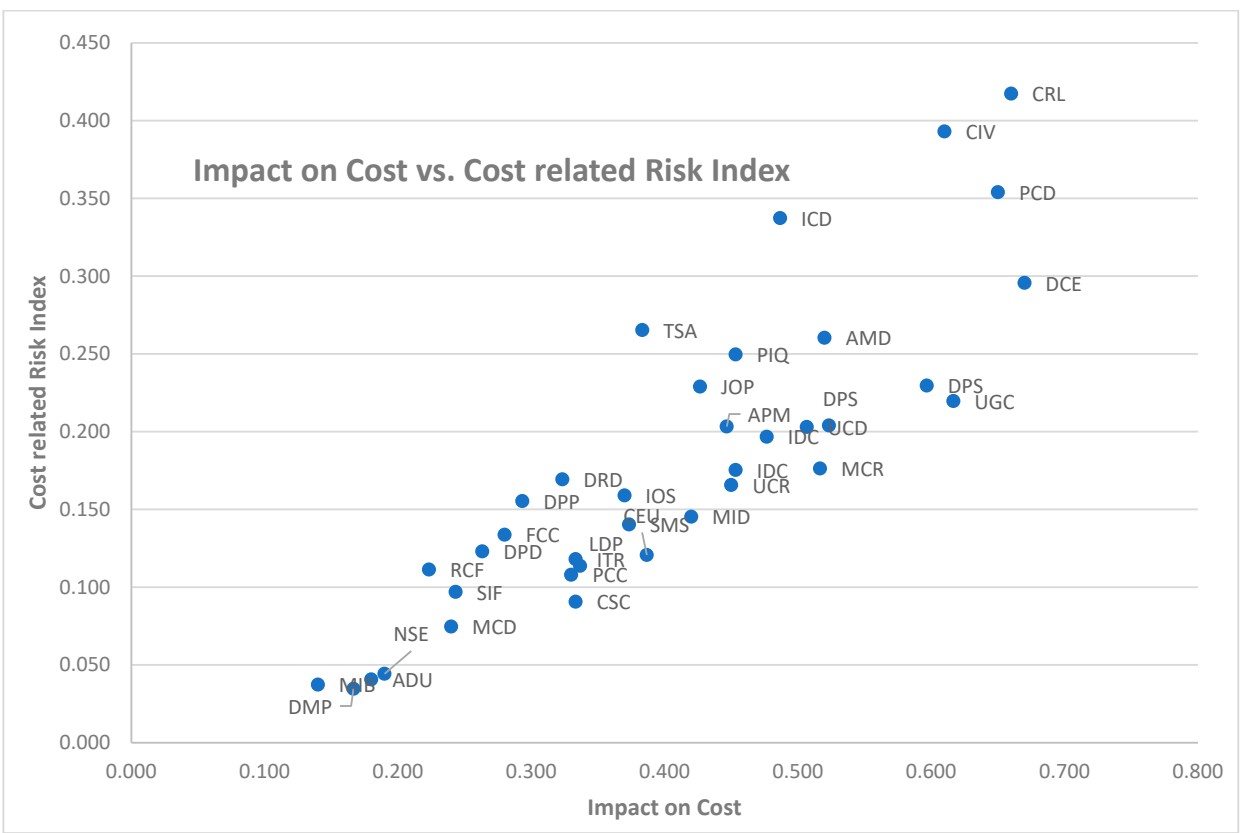

**Figure 11.** Impact of risk on cost vs. cost-related risk index.

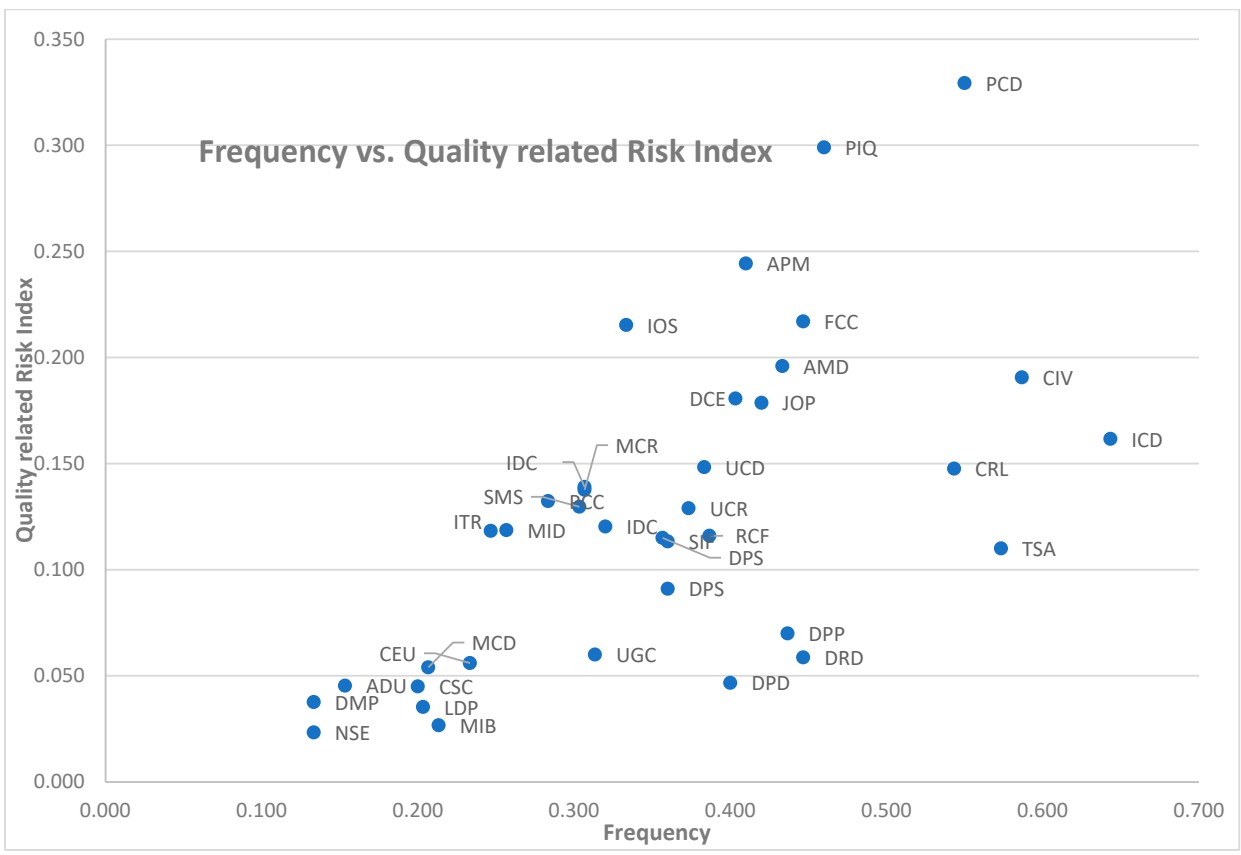

**Figure 12.** Frequency of risk vs. quality-related risk index.

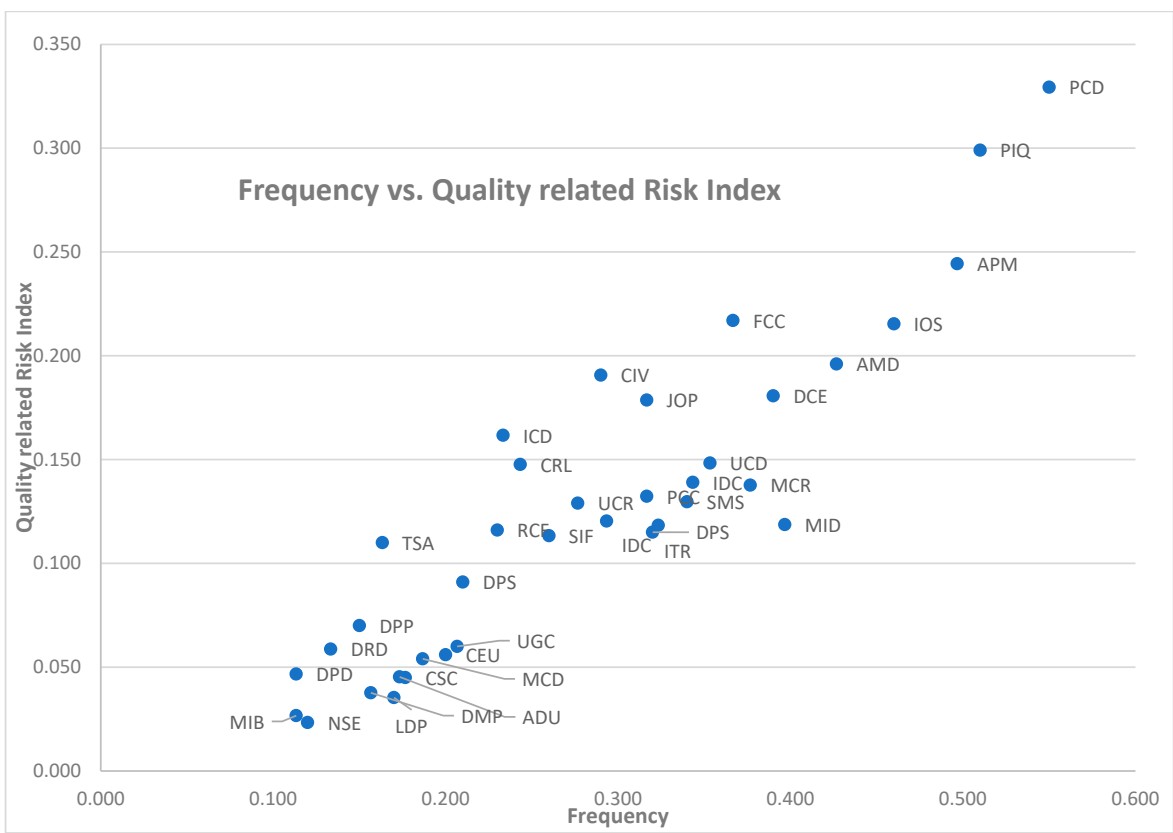

**Figure 13.** Impact of risk on quality vs. quality-related risk index.

### 5.2. Mapping the Root Causes of High-Ranked Risks with BIM-Based Solutions

The root causes of projects with the high-ranked risks mentioned above were further studied to create a BIM-based solution to these problems (Table 5). The experts' opinions on the mapped solution approach validated the claim. The investigations on the possible BIM-based solution approach showed that the BIM could effectively manage 75% of the root causes for the design phase risks affecting the time, cost, and quality of DBB high-rise building projects in the UAE. Further, for the remaining 25%, BIM showed to be effective in managing the outcomes.

**Table 5.** BIM-based solutions for high-ranked identified risks.

| | | A | | | |
|---|---|---|---|---|---|
| **High-Ranked Design Phase-Related Risks** | **Project Objective Impacted by Risk** | | | **Stakeholder** | **Identified Root Causes** |
| | **Time** | **Cost** | **Quality** | | |
| 1  Time spent in the approval process | • | • | | Consultant | Poor implementation of quality assurance/quality control (QA/QC) system (poor reviewing system, poor quality of drawings, mistakes in the drawings) |
| | | | | | Major modifications/comments on the drawings require time from the consultant to rectify the drawings |
| | | | | | Mistakes in the design |
| | | | | | Discrepancies in drawings due to the pressure on the consultant to submit the drawings for approval |
| | | | | | The design was not properly studied by the consultant |
| | | | | | Major changes in the design require time for the consultant to prepare the drawings for approval and for approval authorities to review the drawings |

**Table 5.** *Cont.*

| | A | | | | | |
|---|---|---|---|---|---|---|
| | **High-Ranked Design Phase-Related Risks** | **Project Objective Impacted by Risk** | | | **Stakeholder** | **Identified Root Causes** |
| | | **Time** | **Cost** | **Quality** | | |
| 2 | Complying with new regulations and rules | ● | ● | | Consultant | Mistakes in implementing the new regulations |
| | | | | | | Adopting international codes in the design which are not compatible with the local regulations |
| 3 | Interference by the client in the design process | ● | ● | ● | Client | The client cannot visualize the project/spaces/dimensions/circulation |
| | | | | | | The client has different opinions about the design |
| | | | | | | The client does not have a defined requirements/poor scope definition |
| | | | | | | The client cannot understand the AutoCAD drawings |
| 4 | Client-initiated variations/client requests changes in the design | ● | ● | ● | Client | Clients are not able to explain their requirements |
| | | | | | | Change in real estate market demand |
| | | | | | | Change in the end user requirements |
| 5 | Delay in reviewing and approving design by the client | ● | | | Client | The client cannot understand the 2D drawings |
| | | | | | | The design did not achieve the client's expectations |
| | | | | | | The client cannot judge that the design is the optimal design |
| 6 | Joint ownership of the project | ● | ● | ● | Client | More than one decision maker for the project |
| 7 | Design process suspended by the client | ● | ● | | Client | Not identified |
| 8 | Financial constraints | | | ● | Client | Not identified |
| 9 | Poor coordination between design disciplines | ● | ● | ● | Consultant | Traditional tools are used in coordination between the design disciplines, such as superimposing drawings in AutoCAD, regular meetings between the design disciplines, and hiring a design coordinator. |
| | | | | | | The approval authorities give comments on some drawings which the consultant does not properly coordinate with other drawings. |
| | | | | | | One of the design teams made a change in the design and did not properly coordinate it with other design teams involved in the project. |
| | | | | | | The client requested a change in the design, which is not properly coordinated between the design disciplines. |
| | | | | | | Architectural design is so complicated, and coordination with other design disciplines is difficult using 2D tools. |

**Table 5.** *Cont.*

| A | | | | | |
|---|---|---|---|---|---|
| **High-Ranked Design Phase-Related Risks** | **Project Objective Impacted by Risk** | | | **Stakeholder** | **Identified Root Causes** |
| | **Time** | **Cost** | **Quality** | | |
| 10 Ambiguities and mistakes in drawings and specifications | ● | ● | ● | Consultant | Poor implementation of the QA/QC system |
| | | | | | Traditional tools are used to review the drawings and specifications |
| | | | | | The consultant does not have enough resources to provide more details in the drawings |
| | | | | | The design details are complicated and cannot be presented in 2D. |
| 11 Deficiencies or inaccuracies in cost estimation | | ● | ● | Consultant | The quantity take-off is not accurate |
| | | | | | An approximate method is used in estimating |
| 12 Poor implementation of quality assurance/quality control (QA/QC) system | ● | ● | ● | Consultant | The consultant does not have enough resources to implement a proper QA/Qc system |
| 13 Deficiencies in planning and scheduling of the project | ● | | | Consultant | The consultant does not have a planning department |
| 14 Absence of professional project management | | | ● | Consultant | The consultant does not have a project management department |
| | | | | | The client does not assign a project management consultancy |
| 15 Inappropriate overall organizational structure | | | ● | Consultant | Not identified |
| B | | | | | |
| **High-ranked Design Phase-related Risks** | **Project Objective Impacted by Risk** | | | **Stakeholder** | **Proposed BIM-based Solution** |
| | **Time** | **Cost** | **Quality** | | |
| 1 Time spent in the approval process | ● | ● | | Consultant | Predefined templates, families, annotation, etc., within 3D models provide a sufficient method for minimizing errors |
| | | | | | 3D coordination Clash detection reports Digital design review sessions Extracting drawings from a fully coordinated model |
| | | | | | BIM helps minimize the consequences |
| | | | | | Incorporating changes and finalizing coordination in 3D models |
| 2 Complying with new regulations and rules | ● | ● | | Consultant | BIM helps minimize the consequences |
| | | | | | BIM helps minimize the consequences |
| 3 Interference by the client in the design process | ● | ● | ● | Client | Using visualization technology augmented reality/virtual reality |
| | | | | | Using 3D models and visualization tools improves project team communication and collaboration |

**Table 5.** *Cont.*

| | High-ranked Design Phase-related Risks | Project Objective Impacted by Risk | | | Stakeholder | Proposed BIM-based Solution |
|---|---|---|---|---|---|---|
| | | **Time** | **Cost** | **Quality** | | |
| 4 | Client-initiated variations/client requests changes in the design | ● | ● | ● | Client | Using 3D models and visualization tools improves project team communication and collaboration |
| | | | | | | BIM helps minimize the consequences |
| | | | | | | BIM helps minimize the consequences |
| 5 | Delay in reviewing and approving design by the client | ● | | | Client | Using 3D models and visualization tools Using visualization technology augmented reality/virtual reality |
| 6 | Joint ownership of the project | ● | ● | ● | Client | Using 3D models and visualization tools enables the different decision makers reached to a conclusion |
| 7 | Design process suspended by the client | ● | ● | | Client | BIM helps minimize the consequences |
| 8 | Financial constraints | | ● | | Client | BIM helps minimize the consequences |
| 9 | Poor coordination between design disciplines | ● | ● | ● | Consultant | 3D coordination Clash detection reports Digital design review sessions |
| | | | | | | Using CDE and centralized model working space, with immediate sync, provides proper workflow without any discoordination or missing data between the project team |
| | | | | | | Utilization of digital tools provides an efficient process to facilitate design, fabrication and installation |
| 10 | Ambiguities and mistakes in drawings and specifications | ● | ● | ● | Consultant | Predefined templates, families, annotation, etc., provide sufficient methods for minimizing errors |
| | | | | | | Sections and details can be generated from the 3D model |
| | | | | | | Utilization of digital tools provides an efficient process to facilitate design, fabrication and installation |
| 11 | Deficiencies or inaccuracies in cost estimation | | ● | ● | Consultant | Using accurate MTO extracted from a model |
| 12 | Poor implementation of quality assurance/quality control (QA/QC) system | ● | ● | ● | Consultant | Predefined templates, families, annotation, etc., provide sufficient methods for minimizing errors |
| 13 | Deficiencies in planning and scheduling of the project | ● | | | Consultant | 4D simulation of time |
| 14 | Absence of professional project management | | | ● | Consultant | BIM helps minimize the consequences |
| 15 | Inappropriate overall organizational structure | | | ● | Consultant | BIM helps minimize the consequences |

## 6. Building Information Modelling

BIM is defined in several ways by different authors, but all the definitions focus on the main concept of storing and managing various data produced during the project life cycle in an integral manner to use them in producing the 3D model, 4D time simulation, 5D cost budget, 6D facility management plan, 7D sustainable design, and 8D safety management plan. Building Information Modelling (BIM) is not only a tool, but also a data management platform that requires integration of technologies, people, and processes [33]. BIM has enhanced communication and collaboration between the various project stakeholders and provided a platform where all the stakeholders can not only share information but also retrieve it any time. BIM has improved the design team's capability to fulfil the client's requirements and detect all the physical disagreements in the design that will reduce the possibility to alter the design in the subsequent phases and reduce the cost of design change. The recent advances in BIM are remarkable, especially in the UAE. Dubai has been at the forefront in making the adoption of BIM mandatory in all the projects that involve complex architecture designs, starting from January 2014, for all architectural and MEP works.

## 7. Conclusions

The present research was conducted to determine the impact of key design stage phase-related risks impact on time, quality and cost aspects of high-rise building project sectors delivered under design–bid–build contracts in UAE to assist in developing a BIM-based risk management solution.

A questionnaire was conducted through face-to-face interviews to determine the top 10 risks related to the design phase that have a considerable impact on the time, cost, and quality items of high-rise building projects sectors in the UAE based on their risk significance index score. The risk index was calculated based on the likelihood of occurrence and the impact of risks on time, cost, and quality aspects which were collected through the questionnaire. Further interviews were performed to identify the underlying causes of high-ranked risks and to develop a BIM-based solution for these risks.

The results reveal that the high-ranked risks linked with the design phase and causing a delay in the projects are (ordered from highest to lowest): time spent in the approval process, interference by the client in the design stage, client-initiated modifications/client request for changes in the design, changes in government regulations and laws, poor coordination between design disciplines, deficiencies in planning and scheduling the project, client delays in reviewing and approving the design, joint ownership of the project, poor implementation of quality control/assurance (QC/QA), and ambiguities/imperfections/mistakes in drawings and specifications.

The results revealed that the high-ranked risks associated with the design phase and causing cost overruns in the projects are (ordered from highest to lowest): changes in government regulations and laws, client-initiated modifications/client requests for changes in the design, poor coordination between design disciplines, interference by the client in the design stage process, deficiencies or inaccuracies in cost estimation, time spent in the approval process, ambiguities/imperfections/mistakes in drawings and specifications, poor implementation of quality control/assurance (QC/QA), deficiencies in planning and scheduling the project, and joint ownership of the project.

The results also show that the high-ranked risks linked with the design phase and impacting the quality of the projects are (ordered from highest to lowest); poor coordination between design disciplines, poor implementation of quality control/assurance (QC/QA), absence of professional project management, financial constraints faced by the client, the inappropriate overall organizational structure of companies linked to the project, ambiguities/imperfections/mistakes in drawings and specifications, client-initiated modifications/client request for changes in the design, deficiencies or inaccuracies in cost estimation, joint ownership of the project, and interference by the client in the design process.

The results obtained from the questionnaire were benchmarked with the key risks identified in construction projects in different countries. The findings obtained from the

current study were in line with other research findings, though the majority of this research was carried out during the construction stage of the project's lifecycle and on different types of projects. This gives a strong indication that the Building Information Modelling-based risk management framework that will be developed based on these results can be implemented in a wide range of building projects and other countries.

The root causes of the key design phase-related to risks were identified and used to link the key risks with possible BIM-based solutions. Investigations of the possible BIM-based solution approach showed that BIM could effectively manage 75% of the root causes of the design phase to related risks that have a considerable impact on the time, cost, and quality aspects of high-rise DBB building project sectors in the UAE. Further, for the remaining 25%, BIM was found to be effective in managing the consequences.

Once these risks have been recognized, according to the present study these risks can be prevented during the design stage; also, finding some solution through the BIM will benefit the design and reduce the effect of these risks on time, cost and quality.

**Author Contributions:** Conceptualization, D.B.; methodology, D.B.; validation, R.A.; formal analysis, D.B.; investigation, R.A.; writing—original draft preparation, D.B.; writing—review and editing, R.A. and S.V.; visualization, R.A.; supervision, R.A. and S.V. All authors have read and agreed to the published version of the manuscript.

**Funding:** This research received no external funding.

**Institutional Review Board Statement:** Not applicable.

**Informed Consent Statement:** Not applicable.

**Data Availability Statement:** Not applicable.

**Conflicts of Interest:** The authors declare no conflict of interest.

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
