# Peer review of "Identification of Key Design Phase-Related Risks in DBB Projects in the UAE—Towards Developing a BIM Solution"

_sustainability, doi:10.3390/su15086651_

Round 1
Reviewer 1 Report
The scope of research is clearly presented in the first figure. The literature review was performed correctly, current literature sources were presented. Summing up the article, it can be said that it is a correct analysis of the collected surveys. In my opinion, this is not a work developing the BIM technique. However, it is very important and can shorten the path for other researchers.
Author Response
Please see attached response

Reviewer 2 Report
· 1. Picture quality can be improved to make it more readable. Recheck for labelling. Figure 2 is labelled as table 1. Please recheck labelling of tables and figures. Some tables are not labelled at all. Most figures are wrongly labelled. Line 277, line 314, line 320 and many more are a few examples.
· 2. Some grammatical issues and sentence structure issues like line 39-43, 459-460 and others needs to be corrected.
· 3. Please explain Building Information Modelling in detail and highlight its significance clearly. How has construction industry benefitted from BIM? How can modification of list based on your survey help?
· 4. Also, highlight the significance of this study. Mention a few lines to suggest how study of risks associated with design phase of construction projects can be helpful. Survey is conducted with results. Discussion on the results and the impact of study is missing. Do suggest any limitations, assumptions (if any) and future direction of the study.
Author Response
Please see attached report

Reviewer 3 Report
This study identified the key risks associated with the design phase of multi-story 9 high rise building projects, analyze the root causes, and mapped the root causes of the identified most significant risks against 17 the possible Building Information Modelling based solutions. Some comments may be helpful to impove the quality.
(1) This study focuses on the construction projects in UAE and claim that "The literature review revealed that no study has been conducted so far to determine the key design-phase-related risks that have a distinctive effect on the time, cost, and quality aspects of the construction sectors in the UAE." Hence, are there any specific differences bewteen construction sectors in UAE and other markets? If that is the case, please add more explanation about this in the manuscript.
(2) The manscript only presents the BIM-based solution to the identified risk problems in table 6, there is no detailed explanation about those solutions and why they can work.
Author Response
Please see attached response

Round 2
Reviewer 2 Report
All concerns of the paper are addressed. However, some formatting issues are there.
For example, tables are usually labelled above the table and figures below the illustration. Use same format for labelling on text. Figure 1 used at one place while fig 2 used at other. Full stop is missing on line 457.
Author Response
All the Figures and Tables labelling been corrected according to the respected reviewer comments ,also full stop been added